# Phylogenomic Analysis of Two Co-Circulating Canine Distemper Virus Lineages in Colombia

**DOI:** 10.3390/pathogens9010026

**Published:** 2019-12-27

**Authors:** July Duque-Valencia, Francisco J. Diaz, Julian Ruiz-Saenz

**Affiliations:** 1Grupo de Investigación en Ciencias Animales—GRICA, Facultad de Medicina Veterinaria y Zootecnia, Universidad Cooperativa de Colombia, sede Bucaramanga, Calle 30A # 33-51, 680002 Bucaramanga, Colombia; 2Grupo Inmunovirología, Facultad de Medicina, Universidad de Antioquia, Calle 70 No. 52-21, 050010 Medellín, Colombia

**Keywords:** phylogenomics, reemerging, lineages co-circulation, genome CDV

## Abstract

Canine distemper virus (CDV) is considered a reemerging disease-causing agent in domestic dogs because it presents high divergence among circulating strains worldwide. In Colombia, the South America-3 and South America/North America-4 lineages co-circulate in domestic dogs, both in the Medellin metropolitan area. In this paper, two full CDV genomes from each viral lineage circulating in Medellin were sequenced; we explored the phylogenetic relationship with the available genome sequences; we described the presence of CDV mutations in the South America-3 and South America/North America-4 lineages associated with adaptation to human cells and a crossing of the species barrier and pathogenicity; and we established the evolutionary rates and time of the closest common ancestor for each gene and characterized the presentation of multiple genomic sites by positive selection.

## 1. Introduction

Canine distemper virus (CDV) affects the respiratory, gastrointestinal, integumentary, and neurological systems of domestic and wild dogs [1]. It has been reported in at least 20 families within 7 Orders, thereby demonstrating the great capacity to cross the species barrier [2,3].

CDV belongs to the family *Paramyxoviridae*, genus *Morbillivirus*, which includes other viruses, such as measles, seal distemper, rinderpest, and small ruminant viruses, which are of epidemiological importance for human and animal populations [1,2,3]. CDV is a negative-sense single-stranded RNA virus, and its genome consists of 15,690 nucleotides that include six genes that code for structural proteins: nucleocapsid (N) (encapsulates viral RNA), phosphoprotein (P) (protein cofactor of the viral polymerase), matrix protein (M) (internal layer of viral envelope), fusion protein (F) (fusion between the viral envelope and the membrane of the host), hemagglutinin (H) (cellular tropism), and the large protein (L) (viral polymerase). The P gene also encodes two nonstructural proteins, proteins C and V, which inhibit the response of type I and II interferons [4,5].

For the phylogenetic classification of CDV, the H gene has been used, which shows great genetic diversity with a geographic distribution pattern [6,7]; thus, to date, 19 lineages of CDV have been described worldwide: North America 1–5, Europe/South America 1, South America 2–4, Wild Europe, Arctic-like, Rockborn-like, Africa 1 and 2, Asia 1–4 [8,9,10,11,12,13,14,15].

Due to the increasing need to know the impact of the viral evolution and the crossing of the species barrier on the different genes of CDV, the study of entire genomes by means of phylogenomics has allowed to infer the existing relationships between the strains and their evolutionary history and to clarify some mechanisms of molecular evolution (substitution, insertion, deletion, recombination, duplication, inversion, transposition, and translocation) involved in the evolution of RNA viruses [16]. Additionally, knowledge of genomic characteristics and substitutions involved in the passage between different hosts and between domestic fauna and wildlife has been expanded [10]. Although the availability of the entire CDV genome sequences is still low (83 complete genomes reported in the GenBank) compared with the availability of H gene sequences, most of the lineages described worldwide are well represented and correctly reported [6]. In this study, we characterized the entire genome of two CDV lineages that are co-circulating in Medellin, Colombia.

## 2. Results

### 2.1. Clinical Specimens

The two genomes obtained from clinical samples of ocular and nasal secretion of clinical ill dogs obtained in 2012 and 2017 from the city of Medellin were amplified. The 2012 patient had respiratory and neurological symptoms; the 2017 patient presented respiratory and digestive symptoms, and the vaccination scheme was unknown for both patients.

### 2.2. Phylogenetic Analysis of the Entire Genome

We succeeded in sequencing 15,525 nucleotides of the CDV genome in the two clinical samples, which compared to 56 genomes of the 83 available in the GenBank, because similar genomes belonging to different lineages distributed worldwide were discarded while conducting the preliminary phylogenetic analyses.

Phylogenetic analyses were able to identify 11 of the 19 lineages described worldwide, given that no complete genome sequences of the lineages South America-2, North America-3 and 5, European wild fauna, Rockborn-like, and Asia-4 are reported (Figure 1). When analyzing the tree’s topology, we observe that the vaccine strains North America-1/Asia-3 were used as outgroup. In the first node, there is a bifurcation that gives rise to the North America-2 lineage. On the other branch, we see that a node is formed, which, in turn, bifurcates, giving origin to the South America-3 lineage, and in the other branch, another node is formed that presents a polytomy, giving origin to the South America/North America-4 lineage. On the second branch, a node is formed that bifurcates and gives rise to the Africa-2 lineage. There is also a bifurcation on the other branch of this node that gives origin to lineage Asia-2, and on the other branch, there is another bifurcation that gives rise to the Arctic and Africa-1 lineages. In the third branch of the polytomy, we can observe another node that gives rise to the Asia-1 and Europe/South America-1 lineages.

Phylogenetic trees of each gene were additionally made. It can be seen that the vast majority of N, P, M, and F genes mostly allow for the differentiation of the existing lineages of CDV. However, the topology of the trees of the H gene and genome is different (Figure 2a–e). The L gene does not allow to differentiate the lineages; it presents six bifurcations, in which one presents a polytomy, and the sequences are grouped without keeping the lineages or carrying a geographical pattern, although the strains of the Asia-2 lineage are mostly grouped in the same clade (Figure 2f).

### 2.3. Recombination Analysis

No recombination events were obtained in the circulating lineages in Medellín, however previously reported recombination events were found [17,18].

### 2.4. Analysis of the Amino Acids of CDV

The genome has 5230 amino acids, of which 3064 present variations between the lineages. Of these, 2308 amino acids are common among several lineages, and only 752 are automorphic amino acids.

When evaluating the differences between the amino acid and nucleotide divergence of the different CDV genomes, we found a greater divergence (7–9%) in the distances of amino acids with respect to the distances of nucleotides when comparing the lineages (Table 1). Thus, for example, the amino acid divergence between the North America-1 and South America-3 lineages is 11.2%, whereas the nucleotide distance is 4.9%. By comparing the classification of the genome sequences with respect to the H gene, we can assure that the lineages are preserved by both methodologies. However, if we compare the topologies of the H gene trees and the genome of this dataset, differences can be observed. For example, the South American-3 lineage is as an independent branch in the genome tree, but in the H gene tree, it is grouped in the same clade with the North America-2 lineage. However, if we compare the tables of amino acid distances of both (H gene and genome), we see divergences higher than 4% in the H gene and 8% in the genome, indicating that the sequence classified as South America-3 lineage by both methodologies is correctly classified. If we observe the topology for the South America/North America-4 lineage in the genome tree, we note that it emerges from a bifurcation, whereas in the H gene tree, it emerges from a polytomy between the Europe/South America-1 and North America-2/South America-3 lineages. (Figure 1 and Figure 2, Table 1).

In the analysis of glycosylation sites, we found the same glycosylation sites in the H protein for the South America-3 and South America/North America-4 lineages, that is, 19, 149, 391, and 422, which are sites previously reported in the other lineages. In the F protein, we found the sites 62, 108, 141, 173, and 179 in the South America-3 lineage and 141, 173, and 179 in the South America/North America-4 lineage, which are common sites for other lineages (Table 2).

Conversely, we tried to establish sites that explain the crossing of the species barrier of the South America/North America-4 lineage and South America-3 lineage. It was possible to infer the substitutions presented between the strains obtained from a fox (KJ747371) and the strains obtained from dogs of the South America/North America-4 and South America-3 lineages, respectively. The unique amino acids in the fox strain are described in the sites 259I and 431C in the protein P, 171G in the protein C, 206S in the protein M, 621P in the protein F, and 9D in the protein H.

Furthermore, the CDV genome adapted to human cells (H358) (AB823707) and the genomes of the South America/North America-4 and South America-3 lineages were compared to explore the zoonotic potential, for which the nucleotide sequence of the P gene was edited by adding a guanine between nucleotides 751 and 752 to generate the ORF of protein V [5]. It is worth noting that in reported investigations [19,20], it is said that the quasispecies adapted to human cells had a higher percentage in one nucleotide when compared with the quasispecies isolated from the dog. Table 3 lists the sites identical to the “ancestral” amino acid characterized in the strain not adapted to the H358 cell line, and the sites with the substitutions already present or derived in the lineages of interest are underlined. Therefore, ten sites were found in the two lineages of interest, located in the C, F, and L proteins, which present the substitutions that allow for the adaptation of CDV to human cells.

Finally, we tried to establish sites that may be associated with the pathogenicity of the North/South America-4 and South America-3 lineages; for this, we must bear in mind that the sequences Genome 44Med/CO/2012, KJ47372, and KJ747371 were obtained from animals with neurological symptoms, and Genome 2Med/CO/2017 sequence was obtained from a dog with respiratory and gastrointestinal symptoms, and therefore, the neurological symptoms were constant and the gastrointestinal symptoms were variable among the patients; therefore, we describe the substitutions associated with the gastrointestinal symptoms (Appendix A).

### 2.5. Analysis of Positive Selection

When performing the positive selection analysis, a total of 38 sites were obtained under positive selection in the entire genome, and a total of 1037 sites under negative selection. Genes are under negative selection (Table 4 and Appendix A). Table 4 shows that the proteins with the highest number of sites under positive selection are the P and V proteins, whose function is the evasion of the immune response. Also, we found that the sites with the highest nonsynonymous substitution rate, dN/dS > 10, are P-106, C-3, C-91, C-169, F-53, and H-549 (Appendix A).

### 2.6. The CDV Molecular Clock

The average replacement rate and the time to the most recent common ancestor (tMRCA) were obtained for each gene. We notice that the P gene has the lowest substitution rate, whereas the H gene has the highest substitution rate, and if we average the tMRCA of each gene, we find a tMRCA of 1889 with an interval between 1835 and 1931 (Table 5).

## 3. Discussion

We obtained two complete CDV genomes by using the entire RNA extracted directly from the clinical sample (ocular and nasal secretion) and Sanger sequencing methods without performing viral isolation, due to which we can assure that, with the strategy of working with 15 overlapping 1200-nucleotide fragments, the results are good, as observed in the phylogenetic trees (Figure 1 and Figure 2).

Although the phylogenetic classification of CDV has been established using the H gene, with the availability of genomes of the different lineages we can now verify this classification and, according to our results, the lineages are preserved; however, the topology of the trees is different, probably due to the high evolutionary rates of the H gene compared with the other genes [21]. Although previous studies with a complete genome did not conduct this type of analysis, in our study, we wanted to confirm whether the previously reported lineages were preserved at the genome level. Likewise, while evaluating the phylogenetic trees of each gene, it can be observed that the lineage distribution previously determined is preserved in most of them (Figure 2), except for the L gene. Although this gene is expected to be the most preserved, and therefore, the phylogram presents few clades, in this study, the opposite was noted, possibly due to the recombination events that have been reported in this gene [18]. The demonstrated co-circulation of the two CDV lineages in Medellin opens the possibility of the existence of patients co-infected with the two circulating lineages, however, no evidence of recombination events was found between these two lineages.

The CDV genome has a greater divergence among its amino acids regarding its nucleotides. This is due to the fact that a greater number of nonsynonymous substitutions are occurring, which has as a principle the lack of self-correction of the viral polymerase of RNA viruses [1]. Therefore, attempts have been made to relate these substitutions to vaccine failures, crossing of the species barrier, and virulence. Our study demonstrated the presence of substitutions in sites under positive selection that are linked to these phenomena and are also involved in the evasion of the innate and humoral immune response.

A presentation epitope (sites 444–455) of B lymphocytes (humoral response) was seen in the nucleoprotein, which has been described as highly variable and associated with virulent strains [22]. Our results are consistent with finding substitutions and positive selection pressure in said epitope in the CDV viruses, South America-3, and South America/North America-4 linages, which could potentially explain vaccination failures.

We found sites under positive selection in the P gene (148, 195, 221, and 287) of the South America-3 and North/South America-4 lineages. If we compare these lineages with the vaccine strains, we observe that in site 106, there is a substitution of one glycine for glutamic acid in the North/South America-4 lineage. At site 221, there is a substitution of arginine for glycine in both lineages, whereas there is a substitution of asparagine for serine in both lineages in site 278. These substitutions are present in these sites subjected to positive selection. These sites are also close to or within epitopes of presentation to B lymphocytes reported by Li et al. (2018), and due to the high divergence in these epitopes, the monoclonal antibodies found in this study can differentiate between vaccination and wild-type strains [23]. Considering these results together, it is possible to postulate that changes in this gene also contribute to the low or null production of vaccine antibodies capable of neutralizing wild-type strains (humoral response), as reported for the strains characterized in the United States belonging to the South America/North America-4 Lineage, or may even be one of the explanations for viral immune escape strains [24,25].

Regarding pathogenesis, protein V is involved in suppressing the innate immune response by inhibiting the induction of type I and II interferons and the activity of the NF-κβ complex by the interaction of the Y110 site with the STAT1 molecules, and the W240–W250 region by the binding to STAT2 [4,5,26]. Protein V also inhibits the Mda5 signaling pathway by binding to the R233–E235 sites [26]. Thus, it was determined that protein C suppresses the signaling pathway of INFγ in measles by inhibiting the dimerization of the phosphorylated STAT1 protein, and sites 13, 25, 39, 44, 78, 104, and 168 were found to be related to this activity [27]. On comparing the sites under positive selection of these proteins and those involved in the evasion of the immune response, we find common or very close sites, which shows that this protein influences the increase in biological fitness to adapt to certain environments, such as nerve cells, cells of different species, or even human cells, as previously reported [10,19].

Also, in the fusion protein, a mechanism of resistance to antivirals involving the substitutions I564C, V571C, G572C, and L575C has been described [28]. The same ancestral amino acids I564, V571, and G572 are present in our lineages. Although in this study these sites are not under selection pressure, they should be studied in the lineages of interest for the implementation of antiviral treatment in our environment.

The sites found under positive selection should be considered, paying more attention to sites P-106, C-3, C-91, C-169, F-53, and H-549, which are located in proteins related to the evasion of the innate immune response, transmission, and dissemination to other hosts and crossing the species barrier, respectively [27,29,30], in order to perform in vitro studies to associate them with immunopathogenesis in the two characterized lineages and clarify mechanisms of evasion of the immune response, adaptation to different hosts, and virulence.

We found that the substitution rates per site per year of each gene are high (2.65 × 10^−4^–4.99 × 10^−4^); this is because CDV is a single-stranded RNA virus with a small genome and also presents with recombination, which are characteristics that have been seen in viruses with the highest evolutionary rates [31], the H gene being the one with the highest evolutionary rate, consistent with the reports of Panzera et al. (2015), who consider that in addition to the high evolutionary rates, CDV presents rapid evolution as a result of the intense genetic flow supported in the CDV migration routes and the subsequent influence of particular ecological factors and local selective pressures [10,21].

## 4. Materials and Methods

### 4.1. Type of Study and Ethical Considerations

This is a descriptive and retrospective study, approved by the Ethics Committee for Animal Experimentation of the Cooperative University of Colombia in Bucaramanga. The owners of the dogs signed informed consent forms approved by the ethics committee. In addition, the authors declare that the implementation of this investigation followed all the scientific, technical, and administrative rules for animal research.

### 4.2. Clinical Specimens

The strain MDE-44/12 was obtained from a sampling conducted in the metropolitan area of Medellín, Antioquia, in 2012, which belongs to the South America-3 lineage [8]. The strain MDE-02/17 was obtained from a second sampling conducted in the year 2017, which belongs to the South America/North America-4 lineage [13].

### 4.3. RNA Extraction

The total RNA was extracted from 140 μL of nasal and ocular secretion supernatant using the QIAamp Viral RNA Mini Spin procedure (QIAGEN^®^, Hilden, Germany) according to the manufacturer′s instructions. The RNA quality and quantity was determined by spectrophotometric analysis with a NanoDrop™ One UV-Vis spectrophotometer (Thermo Scientific, Wilmington, DE, USA), and RNA aliquots were stored at −80 °C until they were used.

### 4.4. Synthesis of Complementary DNA (cDNA)

cDNA synthesis was performed using the RevertAid™ Premium cDNA synthesis kit (Thermo Scientific^®^, Glen Burnie, MD, USA) according to the manufacturer’s instructions. Briefly, a denaturing mixture consisting of 1 μL (100 pmol/μL) of random hexamers, 1 μL of dNTP mixture (10 mM), and 13 μL (0.02–4.6 μg) of total RNA was initially denatured at 65 °C for 5 min and immediately incubated on ice. The reverse transcription (RT) mixture solution consisted of 4 μL of reverse transcriptase 5× Buffer and 1 μL of RevertAid™ Premium Enzyme Mix. The RT mixture was added to the denaturation mixture, and the reverse transcription was performed in a total volume of 20 μL in a ProFlex™ polymerase chain reaction (PCR) thermocycler (Applied Biosystems^®^, Foster City, CA, USA) for 10 min at 25 °C, followed by 30 min at 50 °C; the reaction was completed by heating to 85 °C for 5 min. The reaction product was stored at −80 °C until it was used.

### 4.5. PCR and Sequencing

To amplify the complete genome, we used a set of 15 pairs of primers previously reported [32], which have 50 overlapping bp, amplifying a segment of approximately 1000 bp (minus the 5′ and 3′ noncoding ends).

Further, 4 μL of cDNA was added to a PCR reaction mixture consisting of 25 μL of Maxima Hot Start PCR mix (2X), 15 μL of nuclease-free water, and 3 μL (10 μM) of each of the primers. PCR was performed in a ProFlex™ PCR thermocycler (Applied Biosystems^®^) under the following conditions: initial denaturation at 95 °C for 4 min, followed by 35 cycles of denaturation at 95 °C for 30 s, alignment at 60 °C for 60 s, extension at 72 °C for 3 min, and a final extension at 72 °C for 10 min.

After PCR, electrophoresis was performed where 5 μL of the amplicons were added in a 1.5% agarose gel (AGAROSE I™, Amresco, Solon, OH, USA) and were run at 110 V for 60 min. The gels were stained using the EZ-VISION™ DNA immunoassay (Amresco Solom, OH, USA) and observed by transillumination with UV light using the GelDoc™ XR + system with ImageLab™ image acquisition software (Bio-Rad, Hercules, CA, USA). The sizes of the amplification product were estimated using a 100–3000 bp molecular weight scale (GeneRuler™ 100 bp Plus DNA Ladder, Thermo Scientific^®^).

PCR amplicons were sent for purification and sequencing to Macrogen Inc. (Macrogen Inc., Seoul, Korea). The same PCR primers were used for sequencing, and an ABI3711™ automatic sequencer (Macrogen™) was used.

### 4.6. Phylogenetic Analysis

The data obtained from the sequencing were assembled and edited using the SeqMan program (DNAStar Lasergene™ V15.0 software package, Madison, WI, USA). Nucleotide Basic Local Alignment Search Tool was used to explore the similarity of the sequence of the Colombian CDV strains with all CDV sequences available in the National Center for Biotechnology Information nucleotide databases. The alignment of 15,690 nucleotides and their deduced amino acids for the two CDV genome sequences of Colombian strains, together with the 83 genomes available worldwide, was performed with the MEGA 7 program [33]; the ClustalW algorithm was used, and the uncorrected distances (p) of nucleotides and amino acids were calculated.

For phylogenetic analysis of each gene and CDV genome, seven evolutionary models were run (CDV genome = GTR + G + I, L = GTR + G, H = T92 + G, F = T92 + G, M = T92 + G, P = TN93 + G, N = T92 + G) in the MEGA 7 program (Kumar et al., 2016). To make the phylogenetic trees, neighbor-joining distance methods and maximum likelihood (ML) characters were used and confirmed by the Bayes method. A bootstrap of 1000 was used in all methods. The North America-1 lineage sequences were used as an external group in the seven models.

### 4.7. Recombination Analysis

The recombination events were evaluated using RDP4.0 [34] and Simplot v3.5.1 subprograms.

### 4.8. Analysis of the CDV Amino Acids

The amino acid sequences deduced from the genome of the Colombian wild-type CDV strains (5230 aa) were aligned with multiple sequences of CDV genomes from different geographic regions using MEGA 7. The amino acid profile and potential differences with the vaccination strains and wild-type strains of known CDV lineages were assessed. Additionally, the presence of CDV mutations in Colombian lineages associated with adaptation to human cells, crossing of the species barrier, and pathogenicity were evaluated. The prediction of possible N-linked glycosylation sites was made using the NetNGlyc 1.0 package [35].

### 4.9. Analysis of Selection Pressure

To identify the amino acid sites of positive selection in CDV proteins, the relationship of nonsynonymous (dN) to synonymous (dS) substitutions was calculated using ML phylogenetic reconstruction and the general reversible nucleotide substitution model available through the web program Datamonkey [36]. To detect non-neutral selection, the Fast, Unconstrained Bayesian AppRoximation was implemented within the HyPhy software package of the Datamonkey program. The range of significance of the posterior probability is between 0 and 1. In general, the posterior probabilities >0.9 strongly suggest a positive selection, and a Bayes factor of 50 was used to estimate the rates of dN and dS within each codon. The values dN/dS > 1, dN/dS = 1, and dN/dS < 1 were used to define positive selection (adaptive molecular evolution), neutral mutations, and negative selection (purification selection), respectively.

### 4.10. Molecular Clocks

The mean substitution rate (substitutions per site per year) and the time to the most recent common ancestor (tMRCA) of the CDV lineages using a Bayesian approach of the Monte-Carlo Markovian Chain implemented in the BEAUti/BEAST v1.8.4 package were determined for each gene [37]. The analysis was implemented using a strict molecular clock with a constant population size; 3 × 10^7^ generations were run in order to ensure the effective population size >200 for the parameters evaluated using the Tracer v1.7 program [38,39].

## 5. Conclusions

We describe the co-circulation of two CDV lineages in a small area. Also, we characterized the substitutions related to vaccine failures, crossing of the barrier of species, and virulence, which should be kept in mind to elucidate the pathogenesis and control of the CDV in our region. We also confirmed that CDV has multiple sites with positive selection and high evolutionary rates, which is a mechanism involved in the evolution of this infectious agent.

## Figures and Tables

**Figure 1 pathogens-09-00026-f001:**
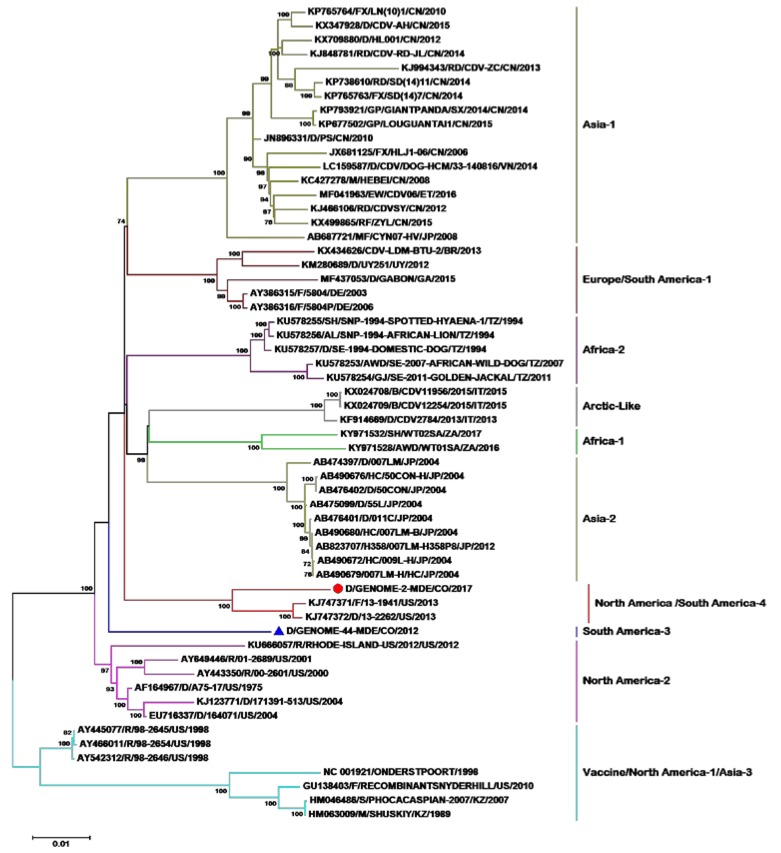
Phylogenetic relationships among 58 CDV (canine distemper virus) strains based on genome sequences. The phylogenetic tree was inferred by the maximum likelihood method using 1000 repetitions. The GenBank access numbers, the species from which each isolate was obtained, the name of the strain, the country of origin, and the year of isolation are indicated on the branch labels, if available. The numbers in the nodes are Bootstrap values for the clade. Abbreviations of animal species: AL: African lion (Panthera leo); AWD: African wild dog (Lycaon pictus); B: Badger (Meles meles); BS: Baikal seal (Pusa sibirica); D: Dog (Canis lupus familiaris); F: Ferret (Mustela putorius furo); FX: Fox (Vulpes urocyon); GJ: Golden jackal (Canisanthus); GP: Giant panda (Ailuropoda melanoleuca); HC: Golden hamster (Mesocricetus auratus); H358: Human lung cells; M: Mink (Neovison vison); MF: crab-eating macaque (Macaca fascicularis); R: Raccoon (Procyon lotor); RD: Raccoon dog (Nyctereutes procyonoides); RF: Red fox (Vulpes vulpes) S: Seal (Phoca vitulina); SH: Spotted hyena (Crocuta crocuta). Country abbreviation: BR: Brazil; CN: China; CO: Colombia; DE: Germany; ET: Ethiopia; GA: Gabon; IT: Italy; JP: Japan; KZ: Kazakhstan; TZ: Tanzania; US: United States; UY: Uruguay; VN: Vietnam; ZA: South Africa.

**Figure 2 pathogens-09-00026-f002:**
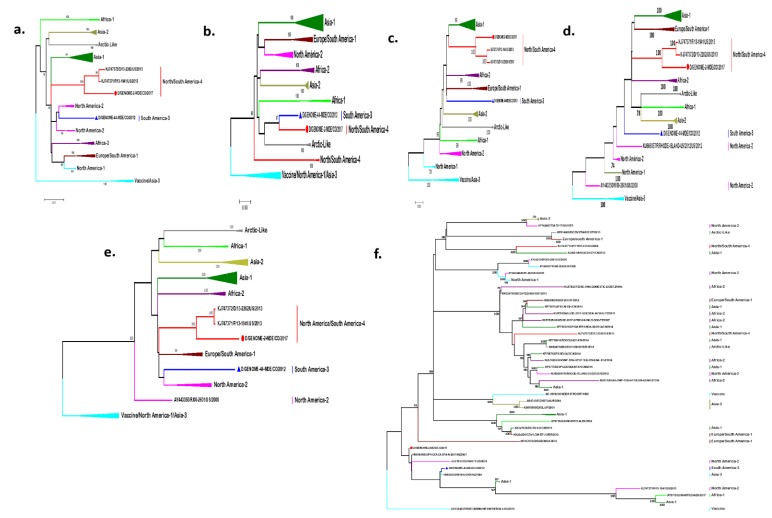
Phylogenetic trees of each CDV gene. They were analyzed by the method of maximum likelihood with a Bootstrap of 1000 replicas. (**a**–**f**): N, P, M, F, H and L genes respectively.

**Table 1 pathogens-09-00026-t001:** Uncorrected distances (p) observed in pairs of amino acid sequences of H genome and gene in CDV lineages.

	SA-4	NA-4	SA-3	VAC	NA-1	ASIA-3	NA-2	EU/SA-1	ARC-L	AFR-1	AFR-2	ASIA-1	ASIA-2
SA-4		0.003	0.004	0.005	0.005	0.005	0.004	0.004	0.005	0.005	0.005	0.005	0.005
		0.007	0.009	0.01	0.011	0.011	0.007	0.008	0.01	0.009	0.009	0.009	0.01
NA-4	0.061		0.004	0.005	0.004	0.005	0.004	0.004	0.005	0.004	0.005	0.004	0.005
	0.031		0.009	0.011	0.011	0.011	0.008	0.008	0.01	0.009	0.009	0.009	0.009
SA-3	0.123	0.121		0.005	0.004	0.005	0.004	0.004	0.005	0.004	0.005	0.004	0.004
	0.06	0.064		0.011	0.012	0.011	0.007	0.008	0.011	0.009	0.01	0.009	0.011
VAC	0.177	0.172	0.174		0.004	0.003	0.005	0.005	0.004	0.005	0.005	0.004	0.005
	0.093	0.095	0.104		0.006	0.006	0.009	0.009	0.011	0.009	0.01	0.01	0.011
NA-1	0.124	0.12	0.112	0.109		0.004	0.003	0.004	0.004	0.004	0.005	0.004	0.005
	0.085	0.088	0.096	0.038		0.007	0.009	0.01	0.011	0.01	0.01	0.01	0.011
ASIA-3	0.173	0.166	0.167	0.057	0.106		0.005	0.005	0.004	0.004	0.005	0.005	0.005
	0.084	0.088	0.096	0.039	0.032		0.01	0.01	0.011	0.01	0.01	0.01	0.011
NA-2	0.099	0.1	0.088	0.149	0.073	0.147		0.003	0.004	0.004	0.004	0.004	0.004
	0.055	0.057	0.056	0.092	0.084	0.083		0.005	0.009	0.008	0.007	0.007	0.009
EU/SA-1	0.13	0.119	0.12	0.17	0.114	0.166	0.097		0.004	0.004	0.004	0.004	0.005
	0.055	0.053	0.056	0.087	0.08	0.077	0.043		0.009	0.008	0.007	0.007	0.009
ARC-L	0.14	0.131	0.135	0.175	0.126	0.167	0.112	0.129		0.004	0.004	0.004	0.005
	0.077	0.071	0.085	0.101	0.092	0.091	0.07	0.066		0.009	0.01	0.01	0.01
AFR-1	0.137	0.129	0.129	0.166	0.126	0.163	0.11	0.13	0.128		0.004	0.004	0.004
	0.062	0.064	0.069	0.092	0.085	0.083	0.062	0.057	0.064		0.009	0.008	0.009
AFR-2	0.132	0.12	0.123	0.169	0.122	0.165	0.102	0.121	0.13	0.129		0.004	0.005
	0.066	0.064	0.073	0.095	0.09	0.089	0.06	0.053	0.08	0.066		0.008	0.01
ASIA-1	0.131	0.118	0.122	0.172	0.122	0.167	0.102	0.118	0.134	0.133	0.123		0.005
	0.065	0.068	0.075	0.097	0.088	0.088	0.061	0.053	0.079	0.067	0.066		0.009
ASIA-2	0.132	0.123	0.126	0.165	0.118	0.161	0.105	0.125	0.121	0.123	0.124	0.128	
	0.071	0.071	0.081	0.102	0.096	0.096	0.07	0.064	0.08	0.068	0.078	0.069	

NA-4: North America-4; SA-4: South America-4; SA-3: South America-3; VAC: Vaccine; NA-1: North America-1, NA-2: North America-2; EU/SA-1: Europe/South America-1; ARC-L: Arctic-like; AFR-1: Africa-1; AFR-2: Africa-2. The values of the genome distances are in boldface, below are the values of the distances of the H gene. The estimated standard error is in blue bold, and it was established with a bootstrap of 1000 replicas.

**Table 2 pathogens-09-00026-t002:** Genome substitutions between vaccination strains and field CDV strains.

Gene	Substitutions
N	T410A, A428T, P448S, H450N, L456F, L456V
P	C40H, C40R, T49A, G50S, N59D, T67A, P97L, P97A, E99D, G106E, E124A, T135A, G137S, R221G, R221E, E233G, L256P, E264G, G270E, N278S, L287P, T426A
M	Not found
F	K3N, K3E, K7E, P23H, R34Q, A35T, Y48H, D49G, T60I, L74S, N82D, Q88H, K96Q, E103K, P107S, I110T, I110V, S616I, A646T
H	H30Q, T56A, D238Y, R241E, R241G, E247K, D329N, H330Q, M342V, K370Q, G376N, N446D, I506T, D531N, N572D, A586T
L	G139D, T149A, N1705K, S1707P, T1708I, S1712L, H2010Q, H2017Y, S2076F

The underlined sites are under positive selection.

**Table 3 pathogens-09-00026-t003:** Substitutions of CDV adapted to Human H358 cells.

Protein	Substitutions
N	L229V, I296M, E467K
P/V	V133A, M267V
V	Y267C
C	L6W, A27V, T30I, T33A, C44S, R47K, R74L, L89P, T93M, A109V, K146R, K154R, Q172R, P173L
M	T84P, F178L, N206D, L329Q
F	C116Y, R331P
H	D540G, M548T
L	D1748N

The sites in which the substitution is already present in the South America-3 and South America/North America-4 lineages are underlined; in the other sites, these two lineages present the “ancestral” amino acid.

**Table 4 pathogens-09-00026-t004:** CDV genome positive and negative selection analysis.

Gene	# Amino Acids	# Positive Selection Sites	# Negative Selection Sites	dN/dS
N	524	2	278	0.0249
P	508	9	90	0.05
V	299	9	33	0.177
C	175	8	12	0.128
M	336	2	155	0.005
F	663	7	262	0.02
H	605	1	207	0.006
L	2185	Not found	Not found	Not estimated

**Table 5 pathogens-09-00026-t005:** Evolutionary model for all CDV genes.

Gene	Substitution Model	Evolution Rate		tMRCA	
		Mean	95% Confidence Interval	Median	95% Confidence Interval
N	T92 + g	3.73 × 10^−4^	2.71 × 10^−4^–4.72 × 10^−4^	1931	1905–1954
P	Tn93 + g	2.65 × 10^−4^	1.69 × 10^−4^–3.61 × 10^−4^	1888	1838–1927
M	T92 + g	3.29 × 10^−4^	1.91 × 10^−4^–4.72 × 10^−4^	1892	1835–1938
F	T92 + g	3.25 × 10^−4^	2.06 × 10^−4^–4.41 × 10^−4^	1835	1766–1891
H	T92 + g	4.99 × 10^−4^	3.87 × 10^−4^–6.13 × 10^−4^	1903	1876–1928
L	Gtr + g	Not calculated	Not calculated	Not calculated	Not calculated

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
