# Peer review of "Phylogenomic Analysis of Two Co-Circulating Canine Distemper Virus Lineages in Colombia"

_pathogens, 2019, doi:10.3390/pathogens9010026_

Round 1
Reviewer 1 Report
This manuscript describes the genetic characterization of two CDV lineages circulating in Colombia. The authors did some phylogenetic analysis using the entire genome and individual gene to investigate the relatedness of these viruses to other known CDV sequences. Overall, the manuscript is adequate, the experimental design is appropriate, and the conclusions are valid. However, there are some grammatical mistakes that need to be fixed for example in line 243 the strand needs to be changed to strain, line 266- delete one of the two “previously,” … etc
I don’t think that the authors need to include table 1 in the Supplementary because the sequences of these primers were previously published
In the results, line 60 the authors mentioned that the phylogenetic analysis was able to identify 11 of the 19 lineages however in figure 1 represents only 10 lineages (not 11).
Figure 2 is not clear (must be in higher resolution).
Line 120-124 in this paragraph the author described the glycosylation site and referred to table 2 at the end of the paragraph, does table 2 show the glycosylation sites??? The title of this table is the genome substitutions between vaccine strains and the field strains. Also, in this table some sites were underlined what does this refer to? Were there differences between the substitutions seen among the vaccine strains and each of the two strains tested in this study?
I do not see where the determined genetic sequences (whole genome sequences) have been uploaded to GenBank or their GenBank accession numbers referenced in the manuscript.
Author Response
Response to Rewiever #1
This manuscript describes the genetic characterization of two CDV lineages circulating in Colombia. The authors did some phylogenetic analysis using the entire genome and individual gene to investigate the relatedness of these viruses to other known CDV sequences. Overall, the manuscript is adequate, the experimental design is appropriate, and the conclusions are valid.
However, there are some grammatical mistakes that need to be fixed for example in line 243 the strand needs to be changed to strain, line 266- delete one of the two “previously,” … etc
R/. We agree to the reviewer. We correct and eliminate the repeated Word.
I don’t think that the authors need to include table 1 in the Supplementary because the sequences of these primers were previously published
R/. We accept the suggestion and eliminate Supp Table 1
In the results, line 60 the authors mentioned that the phylogenetic analysis was able to identify 11 of the 19 lineages however in figure 1 represents only 10 lineages (not 11).
R/: As for the Asia-3 lineage, there is a discussion about its origin due to recombination processes between vaccine strains and circulating wild strains in Asia (Yuan et al. 2017). However when looking at the table of distances between these strains and the strains of the North American lineage -1 the amino acid divergence does not exceed 4%, so in the tree they are grouped in the same clade.
On the other hand, authors such as (Bhatt et al. 2019) report the Asia-3 lineage as a well-defined clada, however, the authors (Si et al 2010) who reported these sequences and (Ke et al. 2015) had shown these sequences in the Asia-2 lineage, our analyzes conclude that Asia-3 is grouped with the strains of the North America-1 lineage, so in the phylogenetic tree only 10 lineages are observed.
Figure 2 is not clear (must be in higher resolution).
R/ We agree to the Reviewer. A HI-Res Image has been upload
Line 120-124 in this paragraph the author described the glycosylation site and referred to table 2 at the end of the paragraph, does table 2 show the glycosylation sites??? The title of this table is the genome substitutions between vaccine strains and the field strains.
Respuesta: We apologize for this mistake. The word Table 2 has been deleted from the paragraph.
Also, in this table some sites were underlined what does this refer to?
R/: The underlined sites refers to the Positive selection. We clarify this in the bottom of the table.
Were there differences between the substitutions seen among the vaccine strains and each of the two strains tested in this study?
R/: Yes, there are differences between each of the field strains and vaccines, some sites that attract attention are included in the table because they are under positive selection.
I do not see where the determined genetic sequences (whole genome sequences) have been uploaded to GenBank or their GenBank accession numbers referenced in the manuscript.
R/ We agree tpo the reviewer. Nevertheless, Some technical issues has been in the BankIT app from the NCBI. Up to date we hadn’t have the final Accession numbers for the full sequences. However the Submission ID for our two genomes are:
SouthAmerica 3: Submission ID: 2293142 South/NorthAmerica: Submission ID: 2292945Reviewer 2 Report
This is a paper describing two full CDV genomes from domestic dogs isolated in Columbia. A detailed analysis has been shown - phylogenetic analysis of the entire genome, recombination analysis, amino acids analysis (showing uncorrected distances, genome substitutions), positive selection analysis, molecular clock. Methods are widely approved and seems to be properly used. The paper is enriched with tables and figures, that helps to follow. Detailed description of methods is helpful for anyone willing to repeat the analysis. Bioinformatic methods also used. This kind of study is important, especially when it shows - as this one- cocirculation of different lineages in a small area, being a sign of possible crossing of the barrier species.This is also a small step forward in describing mechanisms involved in the evolution of the virus.
I would recommend to cite ICTV once describing systematics of the virus.
The paper may be printed in this form.
Author Response
I would recommend to cite ICTV once describing systematics of the virus.
R/: We agree to the reviewer and included the ICTV reference